# Karpinski Score under Digital Investigation: A Fully Automated Segmentation Algorithm to Identify Vascular and Stromal Injury of Donors' Kidneys

**Massimo Salvi [1],\*, Alessandro Mogetta [1], Kristen M. Meiburger [1], Alessandro Gambella [2], Luca Molinaro [3], Antonella Barreca [3], Mauro Papotti [4] and Filippo Molinari [1]**

[1]  Department of Electronics and Telecommunications, Polytechnic of Turin, 10129 Turin, Italy; mogettaalessandro@gmail.com (A.M.); kristen.meiburger@polito.it (K.M.M.); filippo.molinari@polito.it (F.M.)

[2]  Pathology Unit, Department of Medical Sciences, University of Turin, 10126 Turin, Italy; alessandro.gambella@unito.it

[3]  Division of Pathology, A.O.U. Città della Salute e della Scienza Hospital, 10126 Turin, Italy; luca.molinaro@unito.it (L.M.); antonella.barreca@libero.it (A.B.)

[4]  Department of Oncology, University of Turin, 10126 Turin, Italy; mauro.papotti@unito.it

\*  Correspondence: massimo.salvi@polito.it

**Abstract:** In kidney transplantations, the evaluation of the vascular structures and stromal areas is crucial for determining kidney acceptance, which is currently based on the pathologist's visual evaluation. In this context, an accurate assessment of the vascular and stromal injury is fundamental to assessing the nephron status. In the present paper, the authors present a fully automated algorithm, called RENFAST (Rapid EvaluatioN of Fibrosis And vesselS Thickness), for the segmentation of kidney blood vessels and fibrosis in histopathological images. The proposed method employs a novel strategy based on deep learning to accurately segment blood vessels, while interstitial fibrosis is assessed using an adaptive stain separation method. The RENFAST algorithm is developed and tested on 350 periodic acid–Schiff (PAS) images for blood vessel segmentation and on 300 Massone's trichrome (TRIC) stained images for the detection of renal fibrosis. In the TEST set, the algorithm exhibits excellent segmentation performance in both blood vessels (accuracy: 0.8936) and fibrosis (accuracy: 0.9227) and outperforms all the compared methods. To the best of our knowledge, the RENFAST algorithm is the first fully automated method capable of detecting both blood vessels and fibrosis in digital histological images. Being very fast (average computational time 2.91 s), this algorithm paves the way for automated, quantitative, and real-time kidney graft assessments.

**Keywords:** kidney transplantation; digital pathology; deep learning; kidney fibrosis; blood vessel segmentation; convolutional neural networks

## 1. Introduction

Kidney allograft transplant is experiencing a broad revolution, thanks to an increasing understanding of the pathologic mechanisms behind rejection and the introduction of new techniques and procedures for transplants [1]. The primary focus during kidney transplants has always been the identification, assessment, and treatment of allograft rejection. However, recently, a new issue has come to light: a shortage of donor organs. To solve this impasse, selection criteria were revised, leading to the so-called "expanded criteria donor" approach: kidneys that once would have been excluded because of the donors' clinical history or those deriving from deceased patients are nowadays carefully used [2,3].

In this context, the preimplantation evaluation of donors' kidneys has become more and more crucial. The pathologist's challenge is to recognize early signs of degeneration to "predict" the organs' functionality and performance. This analysis, usually based on periodic acid–Schiff (PAS) and trichrome (TRIC) staining, is focused on the glomeruli, tubules, vessels, and cortical parenchyma of the donor kidney, searching for glomerulosclerosis, tubule atrophy, vascular damage, or interstitial fibrotic replacement, respectively (Figure 1). The Karpinski score is then applied to grade the injury of the donor kidney. This score is based on a semiquantitative evaluation of the structures mentioned above. For each of the four compartments (glomeruli, tubules, blood vessels, and cortical parenchyma), the pathologist summarizes the evaluation in a four-grade score, ranging from 0 (absence of injury) to 3 (marked injuries); the total score is expressed out of 12 [4]. Notably, both arteries and arterioles are considered in vascular damage assessment, characterized by progressive thickening of their wall and shrinkage of their lumen. At the same time, the cortical parenchyma could be replaced by fibrous connective tissue [5,6].

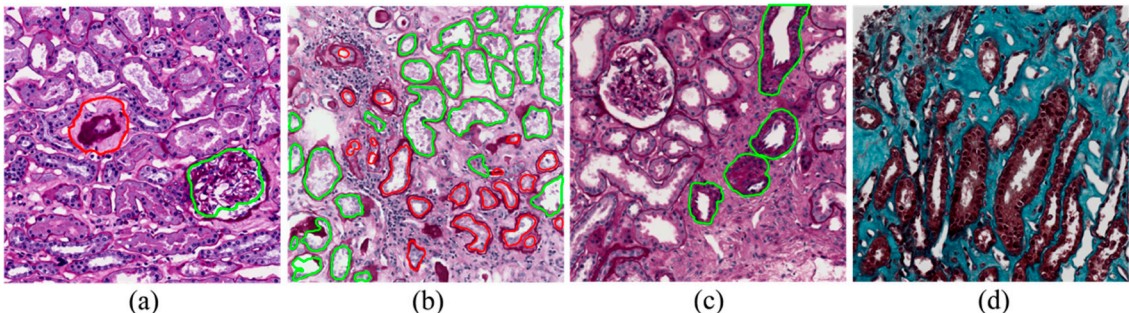

**Figure 1.** Histological features assessed to determine the Karpinski score. (**a**) Glomerulosclerosis: examples of a healthy and sclerotic glomerulus are shown in green and red, respectively; (**b**) Tubular atrophy: healthy and atrophic tubules are highlighted in green and red, respectively; (**c**) Vascular damage: blood vessels are outlined in green; (**d**) Cortical parenchyma: renal fibrosis is represented by the turquoise zone.

The preimplantation kidney evaluation is a delicate, crucial activity for pathology laboratories. It is time-consuming, usually performed with urgency, and has a marked impact on the daily diagnostic routine. Moreover, the evaluation is operator-dependent, with a significant rate of inter-observer difference [7]. In this challenging and evolving panorama, the introduction and application of an automated analysis algorithm would be of compelling importance.

In the last few years, several strategies have been proposed for the segmentation of kidney blood vessels and for the quantification of fibrotic tissue in biopsy images. Bevilacqua et al. [8] employed an artificial neural network (ANN) to detect blood vessels in histological kidney images. Lumen regions were firstly detected by applying fixed thresholding and morphological operators. Seeded region growing was then implemented to extract the membrane all around the segmented objects. Finally, a neural network based on Haralick texture features [9] was used to distinguish between blood vessels and tubular structures. Although well structured, this strategy suffers from several limitations. First, blood vessels with small or absent lumen cannot be segmented using the described approach. In addition, stain variability greatly influences the performance of the region growing, causing imprecise recognition of the blood vessel borders. Finally, the high variability in the shapes, dimensions, and textural characteristics of tubules seriously affects the classification provided by the network. Tey et al. [10] proposed an algorithm for the quantification of interstitial fibrosis (IF) based on color image segmentation and tissue structure identification in biopsy samples stained with Massone's trichrome (TRIC). All the renal structures were identified by employing color space transformations and structural feature extraction from the images. Then, the regions of fibrotic tissue were identified by removing all the non-fibrotic structures from the biopsy tissue area. This approach leads to fast identification of renal fibrotic tissue, but it is not free from limitations. First of

all, there is a loss of information during the color space transformation and, in the presence of high stain variability, the method is not able to correctly classify all the renal structures. Moreover, being based on the identification and subsequent removal of non-fibrotic regions from the tissue, an error in the segmentation of these structures causes inaccurate quantification of interstitial fibrosis. Fu et al. [11] proposed a convolutional neural network (CNN) for fibrotic tissue segmentation in atrial tissue stained with Massone's trichrome. The network, consisting of 11 convolutional layers, was trained on a three-class problem (background vs. fibrosis vs. myocytes), giving the RGB image as input and the corresponding manual mask as the target. This approach provides fast detection of fibrotic areas of the tissue but presents one major limitation: color variability. Stain variations may affect both the training of the network and the correct segmentation of fibrotic tissue, and every mis-segmentation error leads to incorrect detection and quantification of interstitial fibrosis.

In this paper, we present a novel method for the detection of blood vessels and for the quantification of interstitial fibrosis in kidney histological images. To the best of our knowledge, no automated solution has been proposed so far to cope with the issue of stain variability in PAS and TRIC images. Our approach employs a preprocessing stage specifically designed to address the problem of color variability. The proposed algorithm for the segmentation of vascular structures exploits a deep learning approach combined with the detection of cellular structures to accurately segment blood vessels in PAS stained images. Interstitial fibrosis is assessed using an adaptive stain separation method to detect all the fibrotic areas within the histological tissue.

## 2. Materials and Methods

Here we present an automated method called RENFAST (Rapid EvaluatioN of Fibrosis And vesselS Thickness). The RENFAST algorithm is a deep-learning-based method for the segmentation of renal blood vessels and fibrosis. A flowchart of the proposed method is sketched in Figure 2. In the following sections, a detailed description of the algorithm is provided.

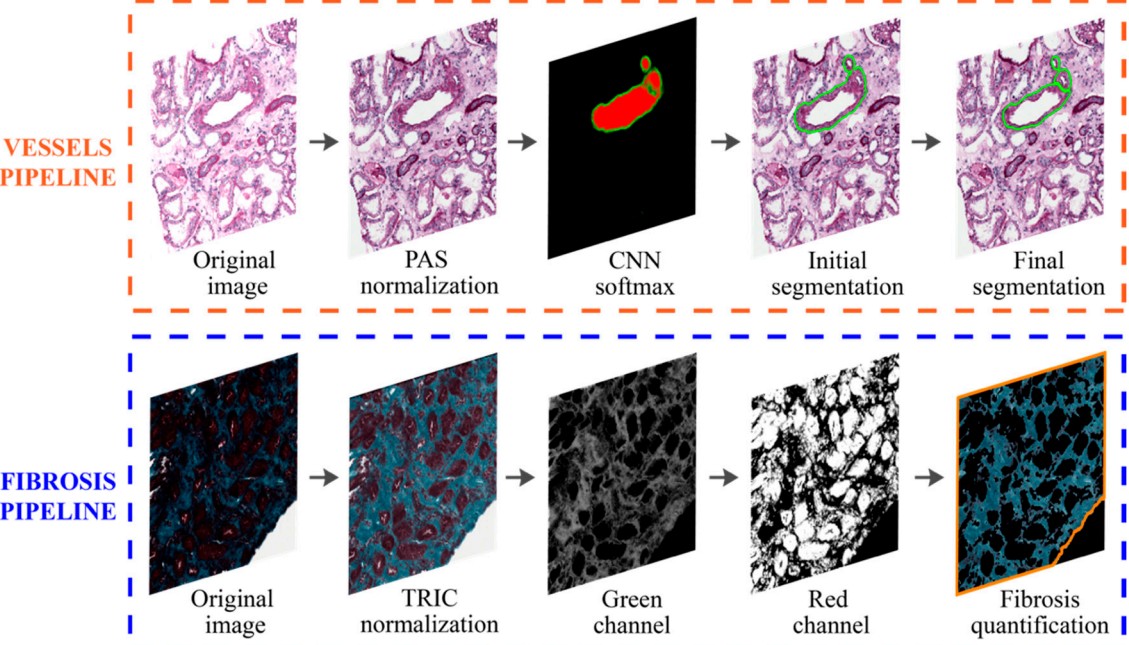

**Figure 2.** Flowchart of the RENFAST (Rapid EvaluatioN of Fibrosis And vesselS Thickness) algorithm for vessel and fibrosis segmentation. The first row illustrates the pipeline for blood vessel detection, while the second row shows the workflow of fibrosis segmentation. After PAS (periodic acid–Schiff) color normalization, blood vessels are detected using a deep learning method (CNN) and ad hoc post-processing. Kidney fibrosis is segmented through TRIC (Massone's trichrome) normalization followed by adaptive stain separation.

### 2.1. Database Description

The whole slide images (WSIs) of kidney biopsy specimens of 65 patients (median age 51 years, range 29–74 years) were used for this work; these were collected at the Division of Pathology, AOU Città della Salute e della Scienza Hospital, Turin, Italy and then anonymized. The pathology laboratory managed the biopsied samples of each kidney according to the kidney transplant biopsy's internal protocol. The tissue was fixed with Serra fixative and then processed in an urgency regimen using a microwave processor or LOGOS J processor (Milestone, Bergamo, Italy). Samples were then paraffin-embedded and serially sectioned (5 μm), mounted onto adhesive slides, and stained with PAS and TRIC. Finally, all the slides produced were scanned with a Hamamatsu NanoZoomer S210 Digital slide scanner (Turin, Italy), providing a magnification of ×100 (conversion factor: 0.934 μm/pixel). For each patient ($n = 65$), an expert pathologist (A.B.) manually extracted 10 images with dimensions of $512 \times 512$ pixels, for a total of 650 images. After consensus, manual annotations of blood vessels and fibrosis were generated by two operators (A.G. and L.M.). Table 1 shows the overall dataset composition. The image dataset, along with the annotations, is available at https://data.mendeley.com/datasets/m2t49zf6xr/1.

**Table 1.** Dataset used in this work.

| Dataset | Subset | Stain | # Patients | # Images |
|---------|--------|-------|------------|----------|
| Vessels | TRAIN | PAS | 30 | 300 |
| | TEST | PAS | 5 | 50 |
| Fibrosis | TRAIN | TRIC | 25 | 250 |
| | TEST | TRIC | 5 | 50 |

### 2.2. Stain Normalization

The proposed algorithm employs a specific preprocessing stage, called stain normalization, to reduce the color variability of the histological samples. Previous studies have shown that stain variability significantly affects the performance of automatic algorithms in digital pathology [12,13]. The procedure of stain normalization allows for transforming a source image $I$ into another image $I_{NORM}$, through the operation $I_{NORM} = f(I, I_{REF})$, where $I_{REF}$ is a reference image and $f(\cdot)$ is the function that applies the color intensities of $I_{REF}$ to the source image [14]. The reference image is chosen by the pathologist as the image with the most optimal tissue staining and visual appearance. For each image of the dataset, the RENFAST algorithm applies the same stain normalization method that we developed in our previous work [15]. First, the image is converted to the optical density space (OD) where the relationship between stain concentration and light intensity is linear. The algorithm then estimates the stain color appearance matrix (W) and the stain density map (H) for both the source and reference images. In order to apply the normalization, the stain density map of the source image is adjusted using the following equation:

$$I_{NORM} = W_{REF} \cdot \frac{H_{SOURCE}}{H_{REF}} \tag{1}$$

where $(\cdot)_{SOURCE}$ and $(\cdot)_{REF}$ denote the source and reference images, respectively. Finally, the normalized image is converted back from the OD space to RGB. Figure 3 illustrates the color normalization process for sample PAS and TRIC images.

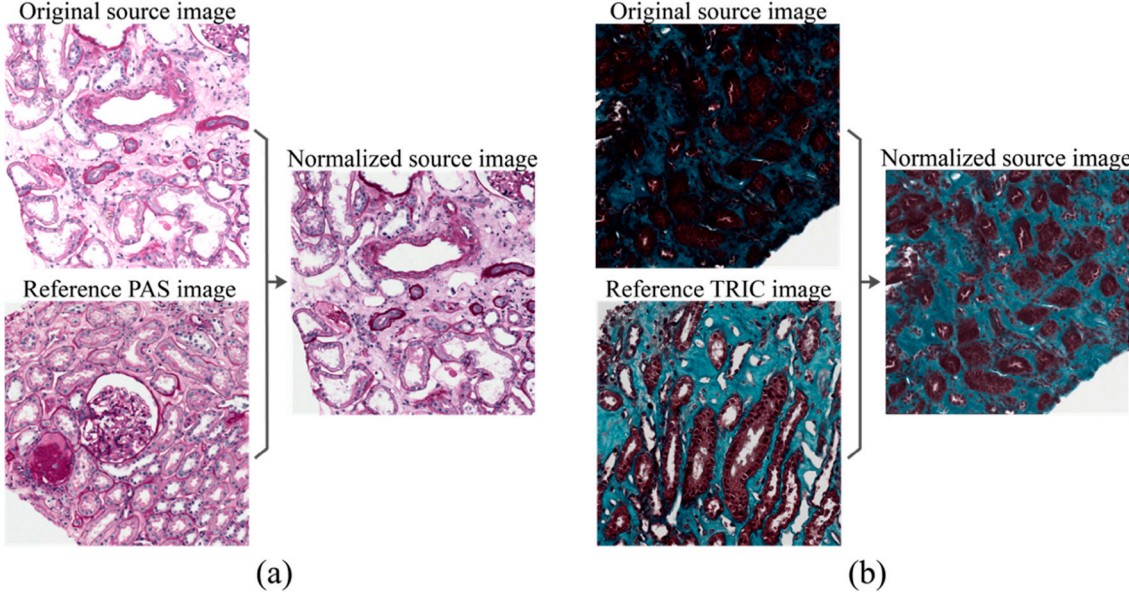

**Figure 3.** Stain normalization performed by the RENFAST algorithm. (**a**) PAS normalization; (**b**) TRIC normalization.

### 2.3. Deep Network Architecture

After stain normalization, the first step performed by the RENFAST algorithm is semantic segmentation using a convolutional neural network (CNN). To perform blood vessel segmentation, a UNET architecture with ResNet34 backbone [16] is employed using the Keras framework. The overall network architecture is shown in Figure 4. This network consists of an encoder structure that downsamples the spatial resolution of the input image through convolutional operations, to obtain a low-resolution feature mapping. These features are then resampled by a decoding structure to obtain a pixel-wise prediction of the same size of the input image. The output of the network is a probability map that assigns to each pixel a probability of belonging to a specific class. The entire network is trained on a three-class problem, giving the $512 \times 512$ RGB images as input and the corresponding labeled masks as the target. In each image of the dataset, pixels are labeled in three classes: (i) background, (ii) blood vessel, and (iii) blood vessel boundaries. To solve the problem of class imbalance, our network's loss function is class-weighted by taking into account how frequently a class occurs in the training set. This means that the least-represented class will have a greater contribution than a more represented one during the weight update. The class weight is computed as follows:

$$f_{classX} = \sum_{i=1}^{N} \frac{\% \, pixel_{classX}}{N} \qquad x = 1, 2, 3 \tag{2}$$

$$class_{WEIGHT} = \frac{median([f_{class1}, \, f_{class2}, \, f_{class3}])}{[f_{class1}, \, f_{class2}, \, f_{class3}]} \tag{3}$$

where $N$ is the total number of images and $f_{classX}$ is the class frequency of generic class X.

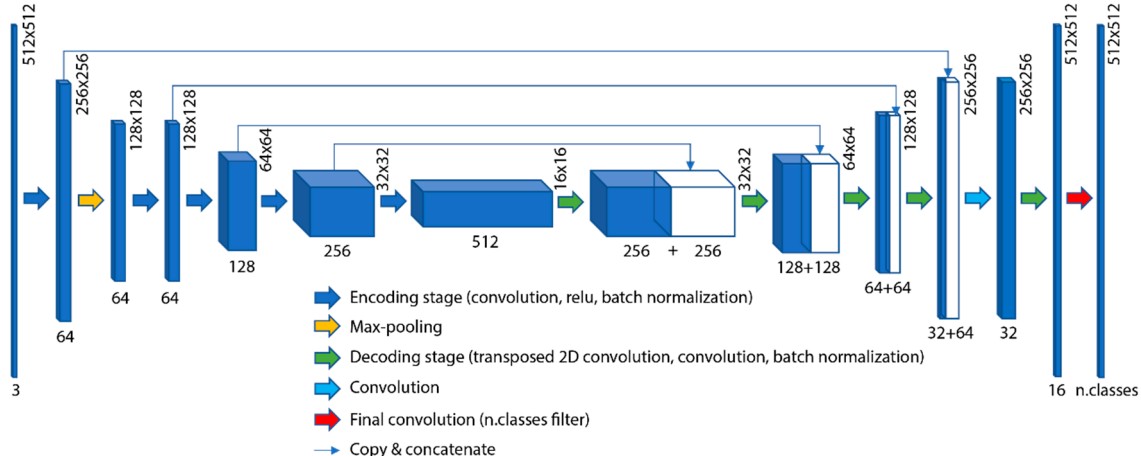

**Figure 4.** Architecture of the deep network employed to perform blood vessel detection. A UNET with ResNet34 backbone was implemented using Keras framework.

The encoding network was pre-trained on ILSVRC 2012 ImageNet [17]. During the training process, only the decoder weights were updated, while the encoder weights were set to non-trainable. This strategy allows for exploiting the knowledge acquired from a previous problem (ImageNet) and using the features learned to solve a new problem (vessel segmentation). This approach is useful both to speed up the training process and to create a robust model even using fewer data. The training data are real-time augmented while passing through the network, applying the same random transformations (rotation, shifting, flipping) both to the input image and to the corresponding encoded mask. Real-time data augmentation allows us to increase the amount of data available without storing the transformed data in memory. This strategy makes the model more robust to slight variations and prevents the network from overfitting.

Our network (Figure 4) was trained on 300 images with a mini-batch size of 32 and categorical cross-entropy as a loss function. The Adam optimization algorithm was employed with an initial learning rate of 0.01. The maximum number of epochs was set to 50, with a validation patience of 10 epochs for early stopping of the training process.

To preserve the information near the boundaries of the image, the RENFAST algorithm applies a specific procedure to build the CNN softmax. Briefly, a mirror border is synthesized in each direction and a sliding window approach is employed to build the probability map. To give the reader the opportunity to observe the entire procedure, we added a detailed description along with a summary figure in Appendix A.

### 2.4. Blood Vessel Detection

Starting from the normalized RGB image (Figure 5a), the RENFAST algorithm applies the deep network described in the previous section. Figure 5b shows the probability map obtained from the CNN, in which the red and green areas represent the pixels inside and on the edge of the blood vessels, respectively. Then, our method detects all the white and nuclear regions within the image. All the unstained structures are segmented by thresholding the grayscale image of the PAS sample, while cell nuclei are detected using the object-based thresholding developed in our previous work [15]. Figure 5c illustrates the segmentation of cellular structures performed by the RENFAST algorithm.

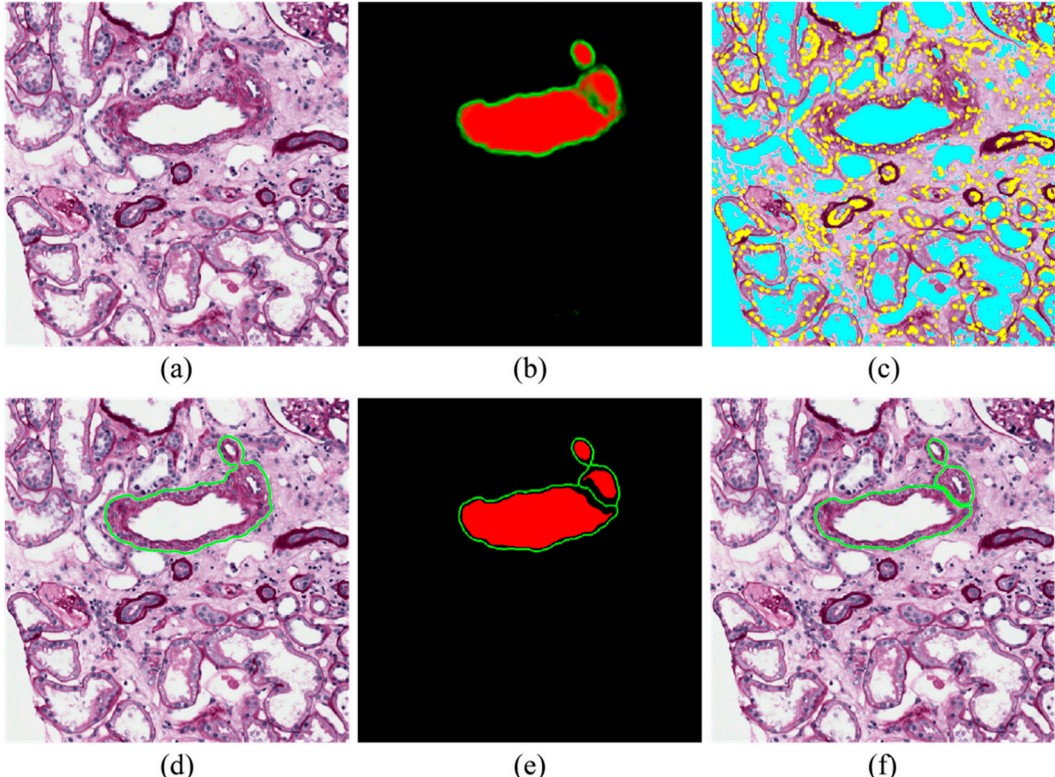

**Figure 5.** Steps performed by RENFAST for blood vessel detection. (**a**) Normalized image; (**b**) CNN probability map; (**c**) Cellular structure detection (yellow: nuclei, cyan: lumen); (**d**) Initial blood vessel segmentation; (**e**) Softmax with high SNR (signal-to-noise ratio); (**f**) Final blood vessel segmentation.

To obtain initial detection of the vascular structures, the probability maps of the regions inside and on the border of the blood vessels are added together and thresholded with a fixed value of 0.35. Then, morphological closing with a disk of 3-pixel radius (equal to 2.80 μm) is carried out to obtain smoother contours. As can be seen from Figure 5d, this strategy leads to accurate detection of the blood vessel boundaries but does not allow the separation of touching structures. To overcome this problem, an additional processing stage is performed to divide clustered blood vessels. The RENFAST algorithm employs a four-step procedure to increase the contrast between each blood vessel's boundary and the background:

1. Inner region mask: thresholding (0.35) and level-set on the probability map of inner regions (red layer);
2. Boundary mask: thresholding (0.35) and level-set on the probability map of boundary regions (green layer);
3. New red layer of the softmax: subtraction of the boundary mask from the inner region mask;
4. New green layer of the softmax: skeleton of the boundary mask.

This procedure generates a softmax with a high SNR (signal-to-noise ratio) where the border of each blood vessel is clearly defined (Figure 5e). Finally, for each connected component of the initial mask (Figure 5d), a simple check is performed: if by subtracting the green layer of the high-SNR softmax (Figure 5e), more than one region is generated, these regions are dilated by 1 pixel and added to the final mask. In this way, the thickness lost during the subtraction is recovered while maintaining the blood vessels' separation. Otherwise, if no additional structure is created with the subtraction, the connected component is inserted directly into the final mask.

The last step of the RENFAST algorithm for vessel segmentation is a structural check on the segmented objects: All the regions with an area less than 180 μm$^2$ are erased as they are too small to be

considered blood vessels. In addition, objects must have at least 2.5% and 5% of the area occupied by lumen and nuclei, respectively. With these structural checks, most of the false positives generated by the CNN are deleted. The final result provided by the proposed algorithm is shown in Figure 5f.

## 2.5. Fibrosis Segmentation

The RENFAST algorithm is also able to quantify interstitial fibrosis in TRIC images. After stain normalization (Section 2.2), our method detects all the uncolored regions to process only TRIC stained structures. The normalized TRIC image is first converted to grayscale and Weiner filtered. The resulting image is then thresholded using a value equal to 90% of the image maximum (Figure 6a). Since fibrosis is characterized by a greenish color, the proposed algorithm applies an adaptive stain separation as described in [15]. Thanks to the stain separation (Figure 6b), it is possible to divide the regions that may manifest fibrosis (green channel) from the structural component (red channel). Segmentation of these two channels is performed using an improved version of the MANA (Multiscale Adaptive Nuclei Analysis) algorithm [18]. After min-max scaling, custom object-based thresholding is applied to the green channel (fibrosis) and red channel obtained in the previous step. For each possible threshold point $T \in [0, 1]$, the RENFAST algorithm computes the following energy function:

$$E(T) = p_0^2 \cdot var_0 \cdot log(var_0) + p_1^2 \cdot var_1 \cdot log(var_1) \qquad (4)$$

where $p_0$ is the probability of having intensity values lower than $T$, $p_1$ is evaluated as $1 - p_0$, while $var_0$ and $var_1$ represent the variances of the probability functions of the two classes $p_0$ and $p_1$. The threshold $T$ associated with the maximum of the energy function $E$ represents the optimal thresholding point. The result of green and red channel segmentation is illustrated in Figure 6c. All remaining pixels not associated with one of the binary masks (white, green, red) are included in the green or red mask based on where they have the highest intensity in the stain separation channel.

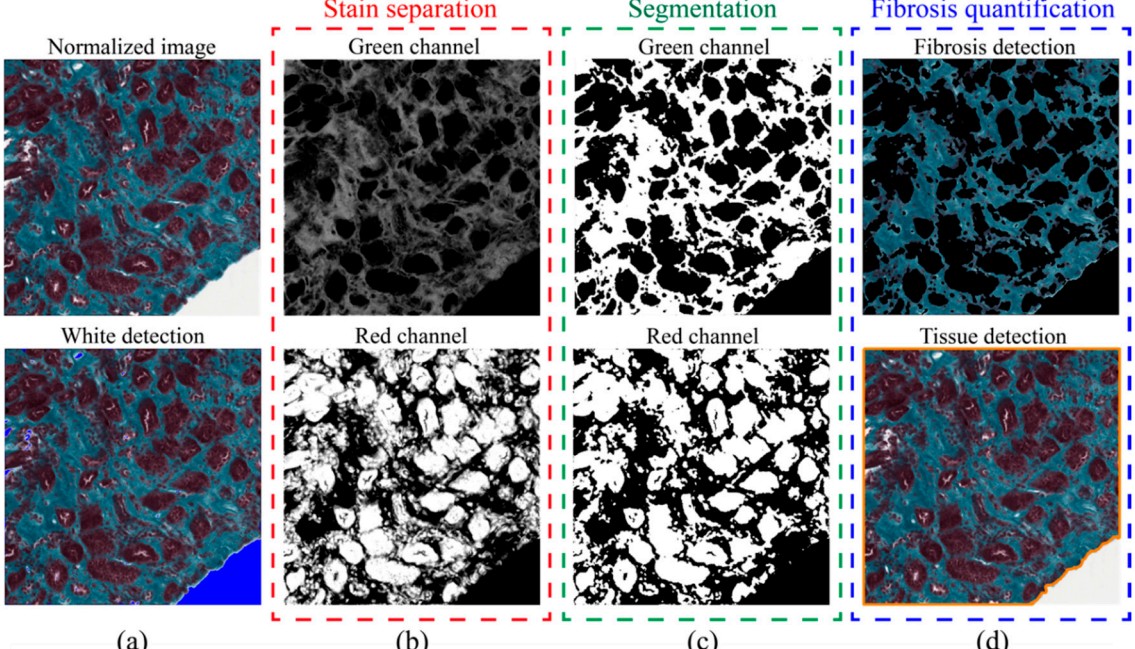

**Figure 6.** Steps performed by RENFAST for fibrosis segmentation. (**a**) Normalized image and white detection (in blue); (**b**) Stain separation between green and red channels; (**c**) Segmentation of green and red channels; (**d**) Fibrosis and tissue detection for interstitial fibrosis quantification.

Finally, the RENFAST algorithm quantifies the interstitial fibrosis as the ratio between the fibrotic area (segmented green channel) and the overall tissue area. Tissue detection is performed using an

RGB high-pass filter [19] where the RGB color of each pixel is treated as a 3D vector. The strength of the edge is defined as the magnitude of the maximum gradient. The raw tissue mask is generated by choosing a threshold equal to 5% of the maximum gradient. Morphological opening with a disk of 4-μm radius is then carried out to obtain the tissue contour (Figure 6d).

*2.6. Performance Metrics*

A comparison between manual and automatic masks was carried out to assess RENFAST's performance in the segmentation of kidney blood vessels and fibrosis. Manual annotations of blood vessels were generated using a custom graphical user interface based on MATLAB. Since fibrosis segmentation can be a long and demanding task, we designed a semi-automatic pipeline to help the pathologist during the generation of the manual mask (Appendix B). Several pixel-based metrics, such as balanced accuracy, precision, recall, and $F1_{SCORE}$, were evaluated for both blood vessel and fibrosis segmentation. Balanced accuracy ($Bal_{ACCURACY}$) is a common metric used in segmentation problems to deal with imbalanced datasets (TP vs. TN). $Bal_{ACCURACY}$ is calculated as the average of the correct predictions of each class individually. Precision is employed to evaluate the false detection of ghost shapes; recall quantifies the missed detection of ground truth objects; and finally, the $F1_{SCORE}$ is defined as the harmonic mean between precision and recall.

Accurate segmentation of blood vessel borders is fundamental for a correct evaluation of vascular damage. For this reason, we also evaluated the Dice coefficient (DSC) and the Hausdorff distance for all the true-positive vascular structures. Specifically, we computed the 95th percentile Hausdorff distance (HD95), which is defined as the maximum distance of a set (manual boundary) to the nearest point in the other set (automatic boundary). This metric is more robust towards a very small subset of outliers because it is based on the calculation of the 95th percentile of distances. During fibrosis assessment, the pathologist computes the ratio between fibrotic tissue and the whole tissue area. For each image, the absolute error (AE) between manual and automatic estimation was calculated as

$$AE = \left| \left( \frac{fibrosis_{AREA}}{tissue_{AREA}} \right)_{MANUAL} - \left( \frac{fibrosis_{AREA}}{tissue_{AREA}} \right)_{RENFAST} \right| \tag{5}$$

where $(\cdot)_{MANUAL}$ and $(\cdot)_{RENFAST}$ denote the manual and the automatic annotations, respectively.

## 3. Results

The automatic results provided by the RENFAST method are compared herein both with manual annotations and with previously published works. For blood vessel segmentation, we compared our algorithm with the one proposed by Bevilacqua et al. [8], while we used the methods published by Tey et al. [10] and Fu et al. [11] as benchmarks for interstitial fibrosis segmentation. As datasets and manual annotations of these works are not publicly available, all the described methods were applied to the same dataset used in this paper. The processing was performed on a custom workstation with a 3.5 GHz 10-core CPU with 64 Gb of RAM (Turin, Italy).

*3.1. Blood Vessel Detection*

Both pixel-based metrics ($Bal_{ACCURACY}$, precision, recall, $F1_{SCORE}$) and object-based metrics (DSC, HD95) were calculated to assess the performance of the RENFAST algorithm. To demonstrate the superiority of our strategy, we also evaluated the results obtained using a simple two-class CNN (background vs. vessel) and a three-class CNN without our post-processing. Tables 2 and 3 summarize the metrics calculated for blood vessel detection.

**Table 2.** Comparison between the RENFAST algorithm and the current state of the art for blood vessel segmentation (pixel-based metrics).

| Method | Subset | Comp. Time (s) | Bal$_{ACCURACY}$ | Precision | Recall | F1$_{SCORE}$ |
|---|---|---|---|---|---|---|
| Bevilacqua et al. [8] | TRAIN | 2.58 ± 1.24 | 0.6845 ± 0.1467 | 0.8618 ± 0.1955 | 0.5115 ± 0.2196 | 0.5996 ± 0.1931 |
| | TEST | 2.64 ± 1.18 | 0.6487 ± 0.1494 | 0.7677 ± 0.2647 | 0.4944 ± 0.2241 | 0.5684 ± 0.2281 |
| Two-class CNN [1] | TRAIN | **0.57 ± 0.11** | 0.8821 ± 0.1116 | 0.9203 ± 0.0945 | 0.8026 ± 0.1630 | 0.8430 ± 0.1242 |
| | TEST | **0.56 ± 0.09** | 0.8116 ± 0.1305 | 0.9308 ± 0.1004 | 0.6923 ± 0.1743 | 0.7741 ± 0.1419 |
| Three-class CNN [2] | TRAIN | 0.74 ± 0.16 | 0.8744 ± 0.0861 | **0.9888 ± 0.0337** | 0.7706 ± 0.1199 | 0.8601 ± 0.0919 |
| | TEST | 0.71 ± 0.18 | 0.8220 ± 0.1075 | **0.9800 ± 0.0800** | 0.6666 ± 0.1837 | 0.7740 ± 0.1597 |
| RENFAST algorithm | TRAIN | 2.67 ± 0.41 | **0.9443 ± 0.0821** | 0.9185 ± 0.0634 | **0.9151 ± 0.0950** | **0.9126 ± 0.0611** |
| | TEST | 2.59 ± 0.53 | **0.8936 ± 0.0969** | 0.9269 ± 0.0845 | **0.8185 ± 0.1344** | **0.8593 ± 0.0858** |

[1] CNN with the same architecture shown in Figure 2 but trained on two classes (background vs. vessel). [2] Same deep network of the RENFAST algorithm but without post-processing (Section 2.4).

**Table 3.** Object-based metrics calculated on detected blood vessels for both the TRAIN and TEST sets.

| Method | Subset | DSC | HD95 (μm) |
|---|---|---|---|
| Bevilacqua et al. [8] | TRAIN | 0.7476 ± 0.1517 | 20.33 ± 21.67 |
| | TEST | 0.7668 ± 0.1381 | 22.31 ± 34.62 |
| Two-class CNN [1] | TRAIN | 0.7447 ± 0.2312 | 21.13 ± 30.59 |
| | TEST | 0.6879 ± 0.2417 | 26.68 ± 36.50 |
| Three-class CNN [2] | TRAIN | 0.7802 ± 0.1777 | 12.02 ± 22.45 |
| | TEST | 0.7483 ± 0.1790 | 9.35 ± 8.84 |
| RENFAST algorithm | TRAIN | **0.8441 ± 0.1762** | **9.78 ± 10.51** |
| | TEST | **0.8358 ± 0.1391** | **6.41 ± 6.25** |

[1] CNN with the same architecture shown in Figure 2 but trained on two classes (background vs. vessel). [2] Same deep network of the RENFAST algorithm but without post-processing (Section 2.4).

Regarding pixel-based metrics, our method achieved the best Bal$_{ACCURACY}$, recall, and F1$_{SCORE}$ for both the TRAIN and TEST sets. A large margin was achieved by RENFAST compared to the state-of-the-art techniques. Even more interesting, the post-processing adopted for blood vessel segmentation allowed a further increase in the overall performance of the single deep network (three-class CNN vs. RENFAST). The combination of the CNN probability map and cellular structure segmentation increased the DSC by up to 14.8% with respect to other methods. The accurate segmentation of blood vessel boundaries is also demonstrated by the lower HD95 value. Figure 7 shows a visual comparison between RENFAST and previously published works. Our approach managed to separate and correctly outline the boundaries of the blood vessels.

*3.2. Fibrosis Segmentation*

The same pixel-based metrics employed in the last section were calculated to evaluate the performance of RENFAST in fibrosis quantification (Table 4). To demonstrate the importance of the stain normalization as a preprocessing step, we also evaluated the performance of our algorithm without normalizing the images ("No norm.").

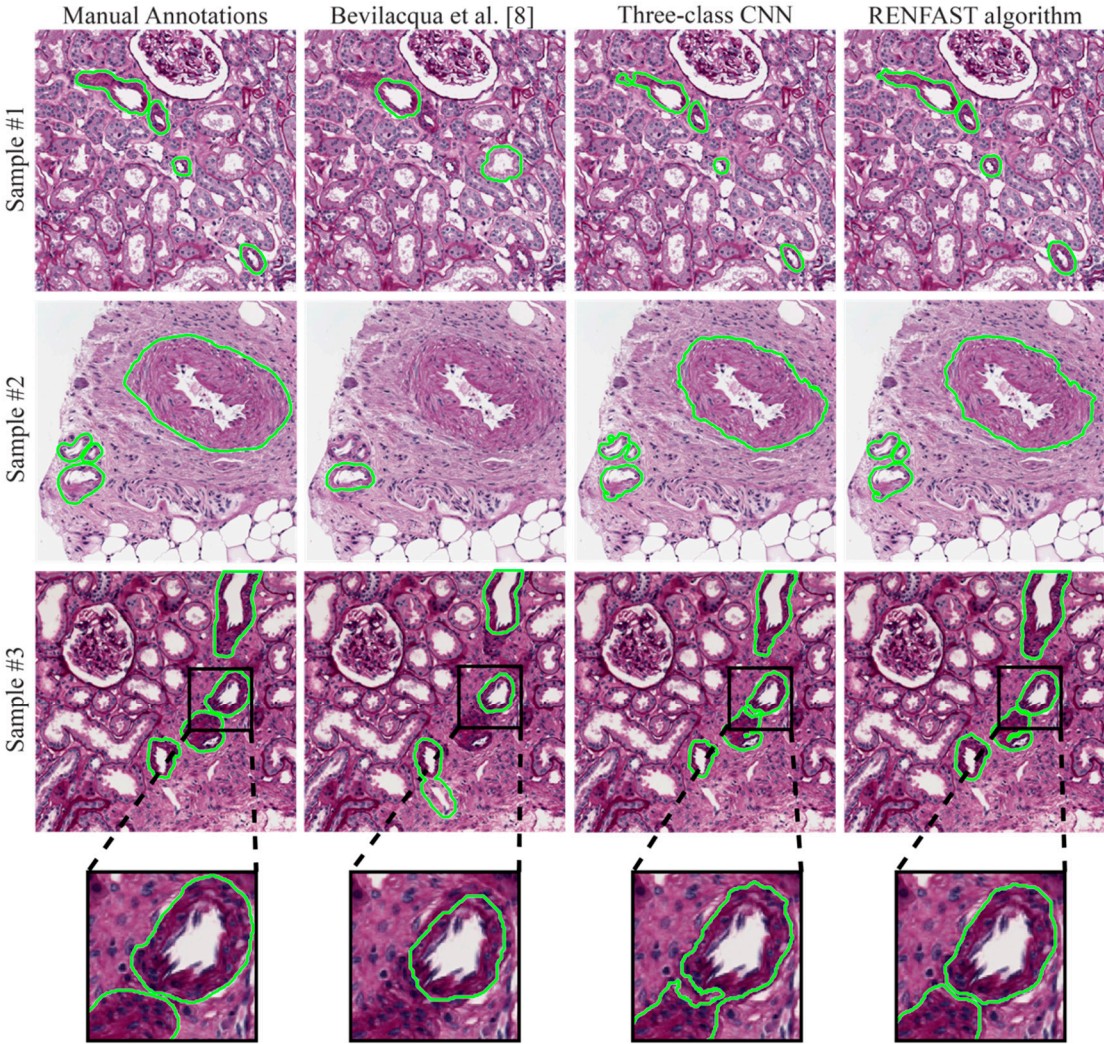

**Figure 7.** Blood vessel detection performed by state-of-the-art methods and the proposed algorithm. Two different samples are displayed in the first rows, while the last row shows a zoom of the segmentation near the blood vessel contour.

**Table 4.** Comparison between the proposed algorithm and the current state of the art for fibrosis segmentation (pixel-based metrics).

| Method | Subset | Comp. Time (s) | Bal$_{ACCURACY}$ | Precision | Recall | F1$_{SCORE}$ |
|---|---|---|---|---|---|---|
| Tey et al. [10] | TRAIN | 0.24 ± 0.04 | 0.8575 ± 0.0374 | 0.7538 ± 0.0780 | 0.8905 ± 0.0744 | 0.8147 ± 0.0515 |
| | TEST | 0.25 ± 0.07 | 0.8604 ± 0.0428 | 0.7512 ± 0.0736 | 0.9055 ± 0.0734 | 0.8166 ± 0.0492 |
| Fu et al. [11] | TRAIN | **0.16 ± 0.06** | 0.8988 ± 0.0660 | 0.8832 ± 0.1072 | 0.8940 ± 0.1469 | 0.8727 ± 0.0896 |
| | TEST | **0.18 ± 0.09** | 0.9159 ± 0.0491 | 0.8783 ± 0.1019 | **0.9239 ± 0.1026** | 0.8911 ± 0.0644 |
| No norm. [1] | TRAIN | 0.17 ± 0.07 | 0.9128 ± 0.0221 | 0.9025 ± 0.0482 | 0.8765 ± 0.0434 | 0.8900 ± 0.0240 |
| | TEST | 0.18 ± 0.11 | 0.9164 ± 0.0247 | 0.9157 ± 0.0304 | 0.8738 ± 0.0499 | 0.8944 ± 0.0277 |
| RENFAST algorithm | TRAIN | 0.27 ± 0.13 | **0.9212 ± 0.0199** | **0.9064 ± 0.0355** | 0.8958 ± 0.0480 | **0.8973 ± 0.0275** |
| | TEST | 0.29 ± 0.14 | **0.9227 ± 0.0222** | **0.9184 ± 0.0313** | 0.8891 ± 0.0482 | **0.9010 ± 0.0246** |

[1] RENFAST algorithm without the stain normalization as preprocessing.

As shown in Table 4, our strategy outperformed all the previously published methods. In addition, the stain normalization (Section 2.2) allowed a further increase in the overall performance of our method (No norm. vs. RENFAST algorithm). Finally, we evaluated the absolute errors (AEs) between the manual and automatic fibrosis quantification (Table 5). In both the TRAIN and TEST datasets, the RENFAST algorithm achieved the lowest average AEs (2.42% and 2.32%), with maximum AEs

of 11.17% and 7.81%, respectively. Specifically, the maximum AE obtained by our method was 3–5 times lower compared to state-of-the-art techniques [10,11]. Figure 8 shows some kidney fibrosis segmentation results.

**Table 5.** Minimum, average, and maximum absolute errors ($AE_{MIN}$, $AE_{MEAN}$, $AE_{MAX}$) between manual and automatic fibrosis quantification.

| Method | Subset | $AE_{MIN}$ (%) | $AE_{MEAN}$ (%) | $AE_{MAX}$ (%) |
|---|---|---|---|---|
| Tey et al. [10] | TRAIN | 0.03 | 8.79 | 42.46 |
| | TEST | 0.59 | 8.73 | 38.41 |
| Fu et al. [11] | TRAIN | 0.01 | 7.81 | 38.62 |
| | TEST | 0.04 | 5.93 | 28.73 |
| No norm. [1] | TRAIN | 0.01 | 2.52 | 11.21 |
| | TEST | 0.05 | 2.50 | 8.29 |
| RENFAST algorithm | TRAIN | **0.01** | **2.42** | **11.17** |
| | TEST | **0.01** | **2.32** | **7.81** |

[1] RENFAST algorithm without the stain normalization as preprocessing.

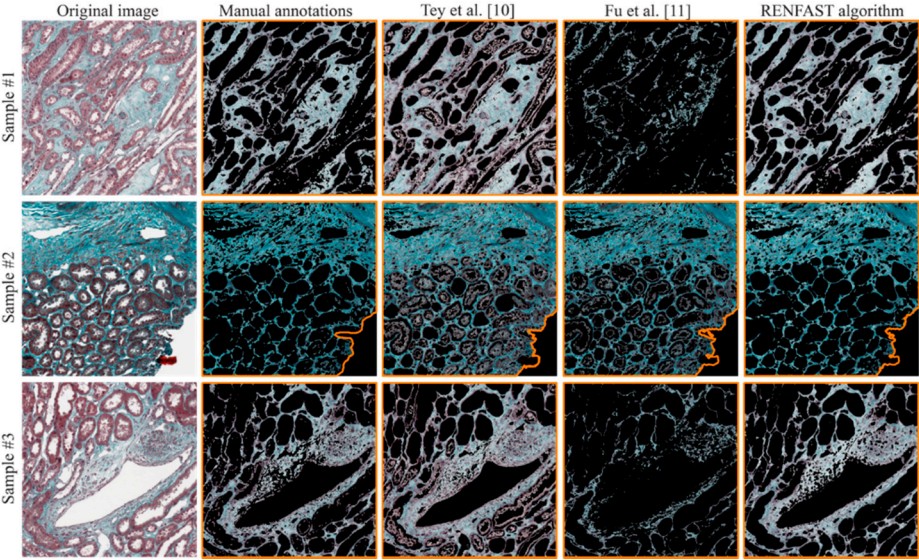

**Figure 8.** Visual performance comparison between previously published papers for fibrosis detection and the RENFAST algorithm. The fibrosis mask is superimposed on the original image, while the tissue contour is highlighted in orange.

### 3.3. Whole Slide Analysis

Since arteriosclerosis and fibrosis are generally assessed on whole slide images (WSIs), we extended our strategy to entire biopsies using a sliding window approach. To evaluate the degree of arterial sclerosis and fibrosis, an expert pathologist takes at least 20 min per patient, while the RENFAST algorithm is able to process the entire WSI in about 2 min. Figure 9 illustrates the results obtained using our algorithm on two different kidney biopsies stained with PAS (vessel detection) and TRIC (fibrosis segmentation). The introduction of an automatic algorithm within the clinical workflow can speed up the diagnostic process and provide more accurate data to assess kidney transplantability.

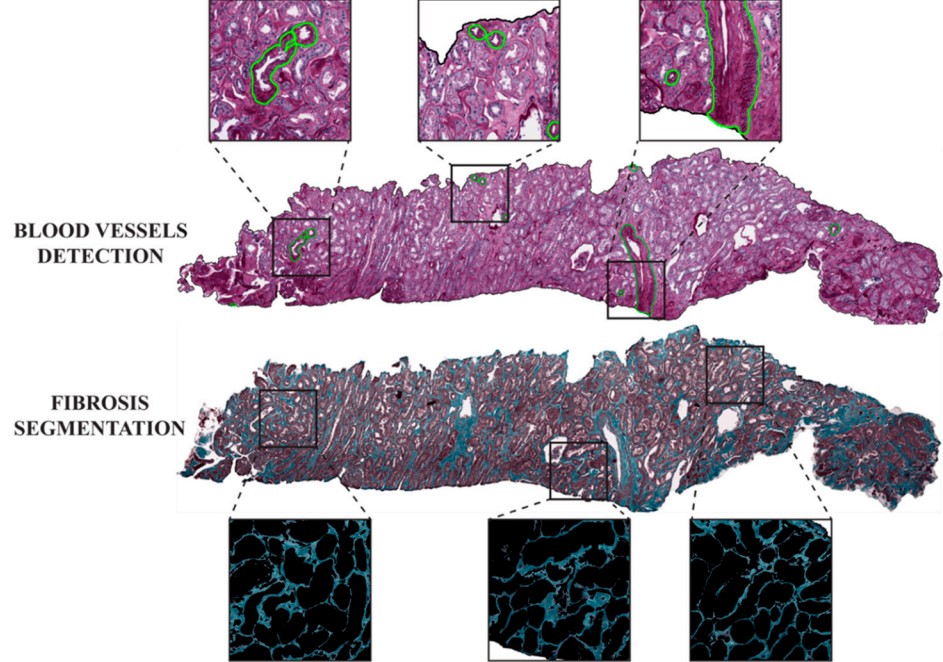

**Figure 9.** The result of RENFAST processing on a whole slide image (WSI). Blood vessels are shown in green in PAS stained WSIs. During the assessment of fibrosis, the connective tissue is segmented by removing all the tubular, vascular, and glomerular structures.

## 4. Discussion and Conclusions

Advances in transplant patient management are steadily increasing with improved clinical data and outcomes, requiring proportional development of the technical procedures routinely applied. However, the histopathological evaluation of preimplantation donor kidney biopsies has not varied, despite the increasing demand for pathology reports.

In this study, we present a fast and accurate method for the segmentation of kidney blood vessels and fibrosis in histological images. The detection of vascular structures and interstitial fibrosis is a real challenge due to the stain variability that affects the PAS and TRIC images, combined with high variation in the shape, size, and internal architecture of the renal structures. Thanks to the stain normalization step, our approach is capable of automatically detecting fibrotic areas and blood vessels in images with different staining intensity. The proposed algorithm was developed and tested on 350 PAS images for blood vessel segmentation and on 300 TRIC stained images for the detection of renal fibrosis. The results were compared with both manual annotations and previously published methods [8,10,11].

In blood vessel detection, the RENFAST algorithm achieved the best $Bal_{ACCURACY}$, recall, and $F1_{SCORE}$ compared to other techniques. More importantly, our strategy obtained the best DSC and HD95 in the segmentation of vessel boundaries (Table 3). This is fundamental as accurate segmentation of the blood vessel borders is mandatory for the correct evaluation of vascular damage. This high performance is mainly due to the combination of CNN segmentation with ad hoc post-processing specifically designed to detect the contour of each blood vessel. By segmenting lumen regions and cell nuclei, the RENFAST algorithm manages to delete almost all the false-positive shapes detected by the CNN. Our strategy is also capable of segmenting small blood vessels and correctly separating touching structures (Figure 7).

On TRIC stained images, the RENFAST algorithm allows us to quantify the interstitial fibrosis. The proposed approach showed high accuracy in segmenting fibrotic tissue and outperformed all the previously published methods (Table 4). Compared with the current state-of-the-art techniques, our method obtained the lowest absolute error (around 2.4%) in the estimation of fibrosis percentage.

In the TEST set, the maximum absolute error of the algorithm was only 7.81%, about 4 times lower with respect to the compared methods. The combination of color normalization and adaptive stain separation allows us to accurately quantify the extent of the fibrotic area.

Although the proposed strategy is fast and robust, it still has some limitations. First of all, the histological images must be acquired at 10× or higher magnification. Using a lower resolution (5× or below), the deep network cannot accurately segment the blood vessels, and cell nucleus segmentation may fail due to the poor quality of the image. Another limitation refers to the WSI application. Nowadays, pathologists evaluate only arteriolar narrowing and interstitial fibrosis in the renal cortex, excluding all structures of the medulla from the evaluation. Our algorithm does not yet include a pipeline for the recognition of the medullary tissue from the cortical tissue on kidney biopsies. However, its potential in assessing vessel and parenchyma injury represents an efficient tool to increase accuracy, reproducibility, and velocity in an increasingly urgent medical setting.

In this study, we presented a simple yet effective pipeline for blood vessel and fibrosis segmentation in kidney histological images. Our research group is currently working on the extension of the RENFAST algorithm to automatically detect the cortical tissue on WSIs and assign a vascular score according to [5]. In the future, we will integrate the assessment of glomerulosclerosis and tubular atrophy within the RENFAST algorithm in order to create the first automated Karpinski scoring system.

**Author Contributions:** Conceptualization, L.M. and F.M.; methodology, M.S.; software, M.S. and A.M.; validation, A.M. and K.M.M.; resources, A.G. and A.B.; data curation, A.G. and L.M.; writing—original draft preparation, M.S.; writing—review and editing, A.M. and K.M.M.; supervision, M.P. and F.M. All authors have read and agreed to the published version of the manuscript.

**Funding:** This research received no external funding.

**Acknowledgments:** The authors would like to acknowledge all the laboratory technicians of the Division of Pathology (Department of Oncology, Turin, Italy) for their help in digitizing histological slides.

**Conflicts of Interest:** The authors declare no conflict of interest.

## Appendix A

During the inference phase, the CNN's probability map could suffer from a lack of information near the edges of the image. To overcome this problem, an *Extended image* is synthesized by padding the original image with mirror reflections of $256 \times 256$ pixels along each direction. As shown in Figure A1, the result of this operation is an RGB image of $1024 \times 1024$ pixels. A sliding window operator with a size of $512 \times 512$ is then passed over the extended image with an overlap of 256 pixels between consecutive windows. The deep network is applied to each $512 \times 512$ window, and only the center of each prediction is kept for the creation of the initial softmax. This operation yields a heat map of size $768 \times 768$ which is further center cropped to obtain the final softmax with the same size as the input image. The final softmax can be considered as an RGB image, where the red layer contains the probability for each pixel of belonging to the "blood vessel" class, while the green layer represents the probability for each pixel of belonging to the "blood vessel boundaries" class.

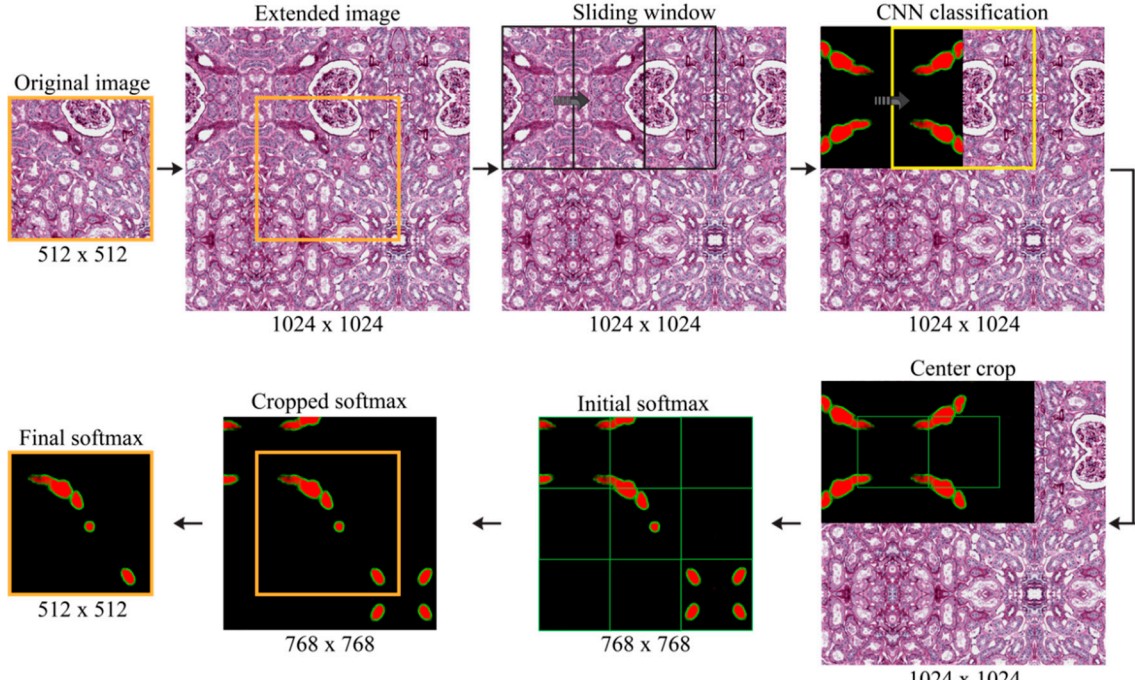

**Figure A1.** Procedure for the creation of the final CNN softmax. The original image is mirrored around the boundaries to obtain the extended image. Then, a sliding window approach is employed to classify each patch, and only the center of each prediction is kept to build the final softmax.

## Appendix B

The semi-automatic pipeline used to generate the manual annotation of fibrotic areas was developed in Fiji [20]. Fiji is a Java-based software product with several plugins that facilitate medical image analysis. The proposed pipeline consists of seven steps: (i) image loading; (ii) manual definition of a ROI (region of interest) for each of the three colors (white, green, red); (iii) RGB color averaging of each ROI to obtain the three stain vectors; (iv) color deconvolution using the stain vectors previously found; (v) manual thresholding on the green channel; (vi) small particle removal; and (vii) complementation of the binary mask.

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
