# Peer review of "Karpinski Score under Digital Investigation: A Fully Automated Segmentation Algorithm to Identify Vascular and Stromal Injury of Donors’ Kidneys"

_electronics, doi:10.3390/electronics9101644_

Round 1

Reviewer 1 Report

The article is very well written. The RENFAST algorithm developed is described in sufficient detail and the the results from the algorithm exhibit substantial improvement over the state-of-art.

Reviewer 2 Report

This manuscript present a segmentation algorithm to detect kidneys’ blood vessels and fibrosis based on histopathological images. The main technique used is a UNET deep learning architecture, which was derived from convolutional neural networks (CNNs) and the ResNet34 backbone.

Overall, the manuscript is well organized and well written. The following suggestions are provided for authors’ further improvement.

  1. The used deep learning backbone network, i.e., the ResNet34, is trained by ImageNet dataset, which consist of images with natural scenes. To the medical images, e.g., the histopathological images used in this research, how to ensure that the trained parameters are still optimal with a small size of new training set?
  2. It would be more convincing if the proposed method is compared with other non-deep-learning based methodologies. After all, image segmentation is not a new applications, and many matured methods have been existing for many years.

Reviewer 3 Report

The authors of the article proposed a segmentation algorithm for the detection of damage degree of the kidney intended to transplantation. They are focused on the analysis of tissue obtained from a kidney biopsy, where a level of vessel damage and degree of fibrotic tissue has to be evaluated. If I had to be honest, I should admit that I'm not an expert from the field of medicine. But, the article is well written and I have been reading it with high interest. I was expecting more image processing, but I appreciate that the authors refer to their previous work. From the results, it’s clear that the proposed RENFAST algorithm achieves better results than state-of-the-art techniques. On the other hand, the proposed algorithm is slightly more time consuming, but I think the longer time period of segmentation is not a big issue. I have no comments on the used scientific methods as well as the article. Maybe a small suggestion, to use more than two pictures of samples in the article and more samples/patients in the evaluation part of the article. I’m glad that I had the opportunity to review this article.
